# The First High-Quality Genome Assembly of Freshwater Pearl Mussel *Sinohyriopsis cumingii*: New Insights into Pearl Biomineralization

**DOI:** 10.3390/ijms25063146

**Published:** 2024-03-09

**Authors:** Zhiyi Bai, Ying Lu, Honghui Hu, Yongbin Yuan, Yalin Li, Xiaojun Liu, Guiling Wang, Dandan Huang, Zhiyan Wang, Yingrui Mao, He Wang, Liangbiao Chen, Jiale Li

**Affiliations:** 1Key Laboratory of Freshwater Aquatic Genetic Resources, Ministry of Agriculture and Rural Affairs, Shanghai Ocean University, Shanghai 201306, China; 2Shanghai Engineering Research Center of Aquaculture, Shanghai Ocean University, Shanghai 201306, China; 3Shanghai Collaborative Innovation Center of Aquatic Animal Breeding and Green Aquaculture, Shanghai Ocean University, Shanghai 201306, China; 4International Research Center for Marine Biosciences, Ministry of Science and Technology, Shanghai Ocean University, Shanghai 201306, China; yinglu@shou.edu.cn (Y.L.);; 5Key Laboratory of Exploration and Utilization of Aquatic Genetic Resources, Ministry of Education, Shanghai Ocean University, Shanghai 201306, China

**Keywords:** pearl, *Sinohyriopsis cumingii*, genome, *fibrillin* gene family, biomineralization

## Abstract

China leads the world in freshwater pearl production, an industry in which the triangle sail mussel (*Sinohyriopsis cumingii*) plays a pivotal role. In this paper, we report a high-quality chromosome-level genome assembly of *S. cumingii* with a size of 2.90 Gb—the largest yet reported among bivalves—and 89.92% anchorage onto 19 linkage groups. The assembled genome has 37,696 protein-coding genes and 50.86% repeat elements. A comparative genomic analysis revealed expansions of 752 gene families, mostly associated with biomineralization, and 237 genes under strong positive selection. Notably, the *fibrillin* gene family exhibited gene family expansion and positive selection simultaneously, and it also exhibited multiple high expressions after mantle implantation by transcriptome analysis. Furthermore, RNA silencing and an in vitro calcium carbonate crystallization assay highlighted the pivotal role played by one fibrillin gene in calcium carbonate deposition and aragonite transformation. This study provides a valuable genomic resource and offers new insights into the mechanism of pearl biomineralization.

## 1. Introduction

Freshwater and marine pearl culture, which has been practiced since ancient times, has evolved into the world’s preeminent aquaculture industry [1]. Since 1984, China has maintained its position as the world’s leading freshwater pearl producer. Today, China is responsible for approximately 90% of total annual global production, with the annual production of freshwater pearls arriving at four hundred tons in the last five years, the value of production exceeding seven billion dollars, and several hundred thousand people working in this sector [2]. Due to its outstanding pearl production performance, characterized by high yields and superior-quality pearl formation (big, round, and colorful), the triangle sail mussel *Sinohyriopsis cumingii* (Lea, 1852) has emerged as the favored species for freshwater pearl production in China. *S. cumingii* inhabits large lakes, rivers, and estuaries across China [3,4]. The abundant genetic resources of *S. cumingii* may be seen as the foundation for its high-quality pearl products. Generally, seawater pearls are of better quality than freshwater pearls in some terms, such as luster, while freshwater pearls have thicker nacre, exhibiting their advantages in yield. Furthermore, the color of freshwater pearls is relatively simple compared with seawater ones, resulting in the low value of freshwater pearls. Improving the quality of pearls is therefore a key concern of the freshwater pearl industry.

Understanding the mechanism of freshwater pearl biomineralization can be seen as the theoretical basis for improving pearl culture technology, including the use of genetic breeding to improve aquacultural growing conditions; however, such improvements also involve considerable challenges. The process of pearl—or shell—formation is a remarkable biomineralization phenomenon involving intricate organic matrices [5,6,7]. This process entails the conversion of ions from the environment into solid minerals, followed by the ordered growth of crystals with the participation of biological cells, ultimately leading to the formation of distinctive biological minerals [8,9]. Numerous organic matrices regulate the deposited calcium carbonate, influencing the arrangement and growth of crystals for nucleation [10]. Gene duplications within carbonic anhydrase (CA), von Willebrand factor A (VWA), and chitin-binding (CB) domain-containing protein families in mollusks have been proposed as key events after their divergence from other lophotrochozoan lineages. These are known to play roles in molluscan shell matrix proteins (SMPs) and in influencing the transition from ancestral exoskeletons to mineralized shells [11]. Notably, the nacre, a convergent carbonate mineral structure, exhibits limited homology, or the absence of similarity, in nacre-associated protein repertoires across bivalves, gastropods, and cephalopods, highlighting evolutionary plasticity [12,13].

Whole-genome sequencing may greatly empower the most fundamental inquiries in biology and evolution; specifically, it may prove instrumental in advancing genetic improvement efforts across various shellfish species. For instance, the oyster (*Crassostrea gigas*) genome has elucidated environmental adaptation and shell formation in bivalves [14]. Two scallop (*Chlamys farreri* and *Patinopecten yessoensis*) genomes have provided insight into the growth and development of bilaterian evolution [15,16]. The pearl oyster (*Pinctada fucata*) genome has shed light on biomineralization in nacre formation [17]. The Nautilus (*Nautilus pompilius*) genome has proven to be a valuable resource in the study of eye evolution and biomineralization in cephalopods [18]. At the present time, genomic resources for freshwater bivalves are limited. Until recently, published genome sequences have been confined to the five freshwater mussels (*Dreissena rostriformis*, *Venustaconcha ellipsiformi*, *Megalonaias nervosa*, *Potamilus streckersoni*, and *Margaritifera margaritifera*) [19,20,21,22,23]. It is especially noteworthy that the whole-genome sequence for the freshwater pearl mussel is still not available. This genome gap has greatly limited our understanding of pearl biomineralization.

In this paper, we report our genome assembly of the triangle sail mussel and a subsequent comparative genome analysis. We present evidence of molecular adaptation and evolution in gene content, which lies behind the biomineralization process. In particular, the research described here shows that the *fibrillin* family contributes especially to the remarkable pearl-production ability of this species. New insights into our understanding of molecular mechanisms are also described. These genomic resources not only advance our comprehension of the mechanism of pearl formation but also lay a solid foundation for future genetic improvements and innovations in culture technology.

## 2. Results

### 2.1. De Novo Sequencing and Genomic Characterization of S. cumingii

High-depth genome sequencing of a single female *S. cumingii* individual was performed to generate a high-quality reference genome. Using the Illumina sequencing technology, 153.49 Gb of Illumina PE150 data, 265.49 Gb of whole-genome shotgun data, and 206.21 Gb of 10× Genomics data were obtained in total (Appendix A). Using Pacbio sequencing, we generated 86.49 Gb of Pacbio HiFi data (Appendix A). After quality control, we obtained 111.35 Gb, 213.70 Gb, 206.20 Gb, and 86.49 Gb of clean data for Illumina PE150, whole-genome shotgun, 10× Genomics, and Pacbio HiFi, respectively. We performed a survey to estimate the genome size and heterozygosity of *S. cumingii* using Illumina (San Diego, CA, USA) PE150 data. The genome had an estimated size of close to 2.91 Gb and exhibited high heterozygosity (0.92%) (Appendix A, Appendix A). The genome assembled with the Illumina and 10× Genomics data consisted of 8268 scaffolds with a scaffold N50 length of 3.19 Mb and a total length of 3.38 Gb, which contained 15,982 contigs with a contig N50 length of 736.17 Kb (Appendix A). The total length of the genome was larger than the estimated size. Hence, we also performed genome assembly using Pacbio HiFi data. Finally, the genome assembled consisted of 1808 contigs with a contig N50 of 5.30 Mb and a total length of 2.90 Gb (Table 1); this was consistent with the estimated genome size.

Using simple sequence repeats (SSRs) and single-nucleotide polymorphisms (SNPs) on two genetic maps, around 2.61 Gb of scaffolds (corresponding to 89.92% of the genome) were anchored onto 19 linkage groups (Figure 1, Appendix A). BUSCO analysis revealed a high level of completeness, identifying 253 (99.22%) complete genes and 1 (0.39%) fragmented gene in 255 BUSCOs in eukaryotes and 920 (96.44%) complete and 22 (2.31%) fragmented genes in a total of 954 BUSCOs in metazoans (Appendix A). The results show that the assembly covered most regions of the genome.

### 2.2. Repeat Element Identification and Genome Annotation

Repeat element analysis showed that the repeat elements identified in *S. cumingii* constituted 50.86% of the whole genome (Figure 1 and Table 2) and specifically included DNA transposons (13.21%), retroelements (12.38%), and unclassified repeat sequences (23.08%) (Table 2). Long terminal repeat (LTR) elements represented the majority of the confirmed interspersed repeats, accounting for 6.86% of the genome (Table 2).

In addition, a total of 37,696 protein-coding genes, with an average CDS length of 1821 bp, were identified (Table 3). All the genes were aligned to public databases, including GO (Appendix A), COG (Appendix A), and KEGG (Appendix A) for functional annotation. A total of 21,246 (60.99%) genes could be mapped to at least one database.

### 2.3. Gene Family Identification and Phylogenetic Analyses

Among the 13 species, a total of 38,275 gene families were identified; of these, 3802 were present in all species, and 1094 were only found in *S. cumingii* (Appendix A). In addition, 130 single-copy orthologous genes were shared among all species. Comparative genomic analysis also showed that 7675 gene families were shared among *S. cumingii*, *P. fucata*, *C. gigas*, *C. farreri*, and *D. rostriformis* (Appendix A). Specifically, *S. cumingii* shared 102 gene families with *P. fucata* and 453 with *D. rostriformis* (Appendix A). According to the results of the KEGG pathway enrichment analysis, the 1094 *S. cumingii*-specific gene families were mainly enriched in 78 KEGG pathways (Appendix A), which were mostly related to the immune system and disease resistance (Appendix A). However, some of them were also enriched in the “ECM-receptor interaction”, “glycosaminoglycan biosynthesis-keratan sulfate”, and “calcium signaling” pathways, which are associated with biomineralization (Appendix A). 

The phylogenetic tree demonstrated that *S. cumingii* is most closely related to *D. rostriformis* and that these two species diverged about 18.5 million years ago (Mya). The bivalve clade, including mussels, scallops, and oysters, is distantly related to *L. gigantea* and *O. bimaculoides*, and it diverged from these two species around 65 Mya and 75 Mya, respectively (Figure 2).

### 2.4. Expansion and Contraction of Gene Families and Positively Selected Genes in the S. cumingii Genome

We performed an expansion and contraction of gene families in the *S. cumingii* genome via a probabilistic model with separated birth and death rates using CAFÉ version 4.2.1 software. In the genome of *S. cumingii*, it was found that 752 and 1193 gene families were significantly expanded and contracted, respectively (*p* < 0.05) (Figure 2). Specifically, the genes in the expanded families were significantly enriched in 84 KEGG pathways (Appendix A), some of which were related to the immune system and disease resistance (Appendix A). In particular, some gene families were enriched in biomineralization-related pathways such as “calcium signaling” and “glycosaminoglycan biosynthesis-chondroitin sulfate/dermatan sulfate” (Appendix A). The genes in the contracted families were significantly enriched in 18 KEGG pathways (Appendix A), which included those related to the biosynthesis and metabolism of some organic compounds (Appendix A). 

Then we used the Yang–Neilsen method in the CodeML program in PAML version 4.7a software to calculate ratios of nonsynonymous to synonymous substitutions and identify the positive selection genes. A total of 237 (1.70%) genes were found to be under positive selection (Appendix A). KEGG pathway enrichment analysis further showed that these genes were significantly enriched in 17 KEGG pathways (Appendix A), which were mostly metabolic pathways (Appendix A). Notably, the characteristic genes (*Hcu_0920.t1* and *Hcu_0920.t33*), annotated as *fibrillin*, were also included in the expanded gene families (OG0000322 and OG0000925). 

### 2.5. Fibrillin Family Genes in S. cumingii’s Genome

A total of 146 fibrillin genes were identified in the genome of *S. cumingii* (Appendix A). According to the topological structure of the phylogenetic tree, the fibrillin proteins of *S. cumingii* were roughly divided into three subgroups, each of which had a certain amount of fibrillins, among which subgroup 1 (yellow) had the largest amount of fibrillin proteins (Appendix A). At the same time, the conserved domain analysis of fibrillin proteins showed that they contain several calcium-binding epidermal growth factor (EGF-CA) domains (Appendix A). RNA-seq showed that *fibrillin* genes generally exhibited diverse tissue expression patterns in *S. cumingii*. The expression patterns of *fibrillin* gene family members were mainly divided into four groups, of which two groups showed a low expression level, and some members were even not expressed in all tissues. The expression levels of the other two groups were relatively high, and most of them were expressed in all tissues. Overall, the expression patterns of the *fibrillin* gene family had poor tissue specificity, with high expression levels in the mantle, ax foot, and heart (Figure 3A). Meanwhile, we analyzed the expression of *fibrillin* genes in sac tissues after mantle implantation (Figure 3B). The expression of *fibrillin* genes was relatively stable at all times and was roughly divided into six groups, with half of the family members showing high expression at all times. Overall, the expression level of the fibrillin gene family was relatively high from day 4 to day 14 after mantle implantation.

### 2.6. Function Analysis of One Key Fibrillin Gene Associated with Biomineralization in Sinohyriopsis cumingii

After the *fibrillin* gene was inhibited by RNA interference, its expression level was significantly downregulated, accounting for 75.1% of the original relative expression (*p* < 0.01, Figure 4A). SEM observations revealed that, in general, nacre growth was dominated by hexagonal aragonite flakes with smooth surfaces and nucleation sites. The disturbed aragonite flakes were irregular, with shapes that ceased to be hexagonal and gradually became round (Figure 4C). Furthermore, the surface of the normal prism layer was flat, and the crystals were closely connected through the organic matrix (Figure 4C). However, the surface of calcium carbonate crystals after interference was rougher, with great size differences, and the organic matrix between crystals appeared hollow (Figure 4C).

The peptide comprising the EGF-CA domain was synthesized, and its effects on CaCO_3_ crystallization were investigated. Crystals in the saturated Ca(HCO_3_)_2_ solution without peptide exhibited smooth rhomboid shapes (Figure 5A,B). Raman spectroscopy showed that characteristic peaks of intensity were at around 267, 1089, and 2952 cm^−1^ (Figure 5C). When crystals were saturated in Ca(HCO_3_)_2_ solution with 40 μg/mL peptide, calcium carbonate tended to be oval-shaped (Figure 5D,E). Raman spectroscopy showed that characteristic peaks of intensity were at approximately 254, 713, 1087, and 2956 cm^−1^ (Figure 5F); this was considered typical of aragonite crystals. Crystals in the saturated Ca(HCO_3_)_2_ solution with 80 μg/mL peptide appeared like radial needles (Figure 5G,H). When crystals were further examined by Raman spectroscopy, the characteristic peaks of intensity were at approximately 257, 713, 1087, and 2960 cm^−1^ (Figure 5I).

## 3. Discussion

In this study, the genome of a female *S. cumingii* individual was subjected to high-depth sequencing to obtain a high-quality genome assembly. The total size of the assembled genome was 2.90 Gb with a contig N50 length of 5.30 Mb, and 89.92% of the genome sequence (2.61 Gb) was anchored onto 19 linkage groups. This represents the largest bivalve genome ever published in a database [14,15,16,17,19,20,21,22,23,24,25,26,27,28,29,30,31,32,33,34,35,36,37,38,39,40,41,42,43,44,45,46]. In contrast to the genome assembly techniques previously used for other freshwater bivalves [19,20,21,22,23], for the present study we used Pacbio HiFi technology, which was more conducive to our genome assembly of *S. cumingii*. Hence, we obtained a higher-quality genome assembly of *S. cumingii*, which exhibited a greater N50 length. The repeat elements identified in *S. cumingii* constituted 50.86% of the whole genome, suggesting a highly dynamic range of repeat content (9.7–62.0%) in mollusks [47]. Furthermore, the proportion of repeat content in *S. cumingii* was found to be higher than in most bivalves, and this may help to explain why *S. cumingii* has the largest genome size among bivalves. The repeat elements included 13.21% of DNA transposons and 12.38% of retroelements. These two repeat elements mediate gene duplication in the genome of the organism [48,49]. Higher proportions of DNA transposons and retroelements indicated that there might be a large number of gene families expanding rapidly during *S. cumingii* genome evolution. Moreover, this genome was predicted to contain a total of 37,681 protein-coding genes. The number of genes in *S. cumingii* is comparable to that of *D. rostriformis* [19]; both species are freshwater bivalves with a reference genome.

Lineage specificity and the expansion of gene families play the most important roles in phenotypic diversity and in evolutionary adaptation to the environment [50]. In the genome of *S. cumingii*, the lineage-specific gene families were significantly enriched in pathways including “calcium signaling”, “glycosaminoglycan biosynthesis-keratan sulfate”, “glycosaminoglycan biosynthesis-chondroitin sulfate/dermatan sulfate”, and “ECM-receptor interaction”. The expanded gene families were mainly enriched in the “calcium signaling” and “glycosaminoglycan biosynthesis-chondroitin sulfate/dermatan sulfate” pathways. The composition of shells and pearls, both of which are products of calcium metabolism, is driven by the deposition of calcium carbonate, a complex process that is highly controlled by calcium signaling [51]. The lineage specificity and expansion of “calcium signaling” pathways confirmed the common sense that the formation of freshwater pearl shells and pearls relies greatly on calcium metabolism. Glycosaminoglycans with sulfonate groups could cooperate with acidic glycoproteins containing carboxyl groups to enrich calcium ions and participate in the nucleation of crystals [5]. Sulfotransferase (CHST), a key enzyme in the biosynthesis of glycosaminoglycans, plays an important role in the process of shell biomineralization by catalyzing the transfer of sulfonic acid groups to produce glycosaminoglycans with a rich negative charge [52]. Hence, it is speculated that the evolutionary impetus for the expansion of the *CHST* family in the genome of *S. cumingii* might also be due to the processes required for rapid biomineralization. Some extracellular matrix (ECM) proteins, such as collagen and VWAP, have been proven to be involved in molluscan biomineralization [17]. In bivalves, ECMs are also important in the composition of the blood circulation system, where amorphous calcium is abundant. The specific families of “ECM-receptor interaction” pathways may indicate that matrix proteins enter the shell through the circulatory system, or through granulocytes and exosomes, to participate in mineralization, thus supporting the cellular model of Mollusca mineralization [14,53,54]. However, some biomineral-related gene families previously found to be expanded in the pearl oyster genome [17], such as chitin synthases (CHSs), *chitinase*, VWA-containing proteins (VWAPs), and tyrosinase (Tyr), were not expanded in *S. cumingii*, suggesting that a diversity of mechanisms is involved in pearl formation in both seawater and freshwater mussels.

Through comparative genome analysis, we identified a significantly expanded family that was annotated as *fibrillin*. Fibrillins are crucial components of extracellular matrices and play a role in providing structural support [55,56]. We found more *fibrillin* genes in the genomes of *S. cumingii* and *P. fucata* than in those of *O. bimaculoides*, a cephalopod without mineralized shells; this implies that these genes play a role in shell formation. Fibrillin proteins contain many EGF-CA domains, which have been confirmed to exhibit a high affinity for calcium [57,58,59,60]. Based on RNA-seq data, the constant expression indicated that fibrillin genes continuously form microfibrils and then affect chitin network formation to play important roles in organic scaffold construction following crystal deposition [61], thus contributing to biomineralization during pearl formation. In the expanded *fibrillin* gene family, one gene was under positive selection, and its expression represented upregulation during pearl formation; this was used for preliminary functional analysis. We discovered that silencing the *fibrillin* gene led to irregular growth in mineralization layers, which has been previously observed in other shell matrix proteins [62,63]. Through an in vitro crystallization assay using the EGF-CA peptide, we observed that this peptide significantly altered the morphology of prismatic calcite. Despite this, the peptide with a higher concentration also induced the formation of spiculate crystals, commonly known as vaterite. Raman spectroscopy verified that these crystals were similar to calcite. The vaterite constitutes an unstable transitional phase in spherical shell formation and can be easily transferred to aragonite or calcite. A previous study suggested that the soluble organic matrix induces vaterite formation in the otolith [64]. Several shell matrix proteins in seawater pearl oysters have also been found to induce and stabilize vaterite formation [63,65,66]. Our findings suggest that *fibrillin* genes are crucial for the deposition of calcium carbonate and the formation of amorphous crystals during the initial stages of biomineralization. Indeed, the excellent biomineralization ability of the mussel probably results from the massive expansion of the *fibrillin* gene family during the evolution of *S. cumingii*.

## 4. Materials and Methods

### 4.1. Sample Preparation and Sequencing

A healthy three-year-old female triangle sail mussel, *S. cumingii* (Figure 6), was sampled at the Chongming aquaculture base of Shanghai Ocean University (Chongming District, Shanghai, China) for genome sequencing. Genomic DNA was extracted from the mussel using the TIANamp Marine Animals DNA Kit (TIANGEN, Beijing, China) following the manufacturer’s instructions. The genome sequencing of *S. cumingii* involved the creation of three distinct sequencing libraries, following established procedures: one library with a 350 bp insert size; another with 20 kb inserts; and a third with inserts ranging from 15 to 18 kb. The 350 bp insert-size library underwent sequencing using the whole-genome shotgun method on the Illumina platform. The library with inserts exceeding 20 kb was sequenced using the 10× Genomics sequencing platform. Finally, the library with insert sizes ranging from 15 to 18 kb was sequenced on the Pacbio Sequel II/IIe platform. Genome Size Estimation and de novo Genome Assembly.

The raw data of Illumina and 10× Genomics reads were filtered using fastp version version 0.23.2 software (https://github.com/OpenGene/fastp, accessed on 10 June 2023) to remove the low-quality sequences and adapter-contained reads. Further, ccs version 6.4.0 software (https://github.com/PacificBiosciences/ccs, accessed on 15 July 2023) was used to perform the quality control of Pacbio HiFi raw reads. The genome size was estimated by calculating the rate of K-mer number and peak depth using jellyfish version 2.2.7 [67] software. The Illumina and 10× Genomics reads were assembled into scaffolds using Supernova version 2.1.1 [68]; then, the scaffolds were anchored onto the chromosomes using data obtained from two *S. cumingii* genetic maps containing 492 SSRs and 4330 SNPs [69,70]. The Pacbio HiFi reads were also used for haploid genome assembly using HiFiasm version 0.16.1 with default parameters [71]. We then used the newly developed program Khaper to select primary contigs and filter redundant sequences from the initial assembly [72]. Finally, the filtering contigs were also anchored onto the chromosomes using the molecular markers in the genetic maps. The Benchmarking Universal Single-Copy Orthologs (BUSCO version 5.22) program [73] was used to evaluate the assembly’s completeness by estimating the core genes based on the eukaryote odb10 and metazoan odb10 databases.

### 4.2. Repetitive Sequence Identification and Genome Annotation

Repetitive sequences in the genome assembly were identified through both de novo identification and homologous sequence alignments. RepeatModule version 1.0.11 was used to build a repeat library via de novo identification with default parameters. The Repbase database [74] was analyzed in RepeatMasker version 4.0.9 to identify repetitive sequences based on homology [75].

Based on the de novo methods, gene models were obtained using EVidenceModeler version 1.1.1 [76]. The predicted protein-coding genes were aligned with public databases, including the Clusters of Orthologous Groups (COGs) database [77] and the Kyoto Encyclopedia of Genes and Genomes (KEGG) [78], using BLASTP version 2.15.0 with a default E-value threshold (1 × 10^−5^). The Gene Ontology (GO) database was also used for function annotation in Blast2GO version 2.5 [79].

### 4.3. Gene Family Identification and Phylogenetic Analysis

The protein sequence sets of 9 bivalves and 3 other invertebrate species were retrieved from the NCBI database for gene family analysis; these included *Bathymodiolus platifrons*, *Chlamys farreri*, *Crassotrea gigas*, *C. virginica*, *D. rostriformis*, *Helobdella robusta*, *Lingula anatina*, *Lottia gigantea*, *Mytilus galloprovincialis*, *Modiolus philippinarum*, *Octopus bimaculoides*, and *P. fucata*. OrthoFinder version 2.3.3 [80] was used to assign gene family clusters with default parameters. The enrichment of *S. cumingii*-specific gene families in KEGG pathways was also analyzed using KOBAS version 1.2.0 [81].

A phylogenetic tree was inferred using the 130 shared single-copy orthologs identified in OrthoFinder. Multiple alignments of each protein sequence were performed using MUSCLE version 3.8.31 [82], and the conserved blocks obtained from the alignments were selected in Gblocks version 0.91b [83]. All alignments were subsequently concatenated to form a super-protein sequence and used to generate the maximum likelihood (ML) phylogenetic tree in RAxML version 8.2.12 [84] with 1000 bootstrap replicates. Divergence times were estimated using the MCMCTree program in the PAML package version 4.7a [85] based on the topological structure and comparison matrix. The reference divergence time was obtained from the TimeTree database [86].

### 4.4. Gene Family Expansion and Contraction, and Positive Selection Analysis

Expansion and contraction of the conserved homolog clusters were determined in CAFE version 4.2.1 [87] using a probabilistic model with separated birth and death rates and a *p*-value threshold of 0.01. Conserved coding DNA sequence (CDS) alignments of each single-copy gene family were extracted for further identification of positively selected genes. The ratios of nonsynonymous to synonymous substitutions (Ka/Ks) were estimated for each single-copy orthologous gene using the Yang–Neilsen method in the CodeML program [88] with the branch-site model implemented in PAML version 4.7a. A likelihood ratio test was conducted, and a false discovery rate correction was performed for multiple comparisons. Genes with a rate of Ka/Ks > 1 and a corrected *p*-value < 0.05 were defined as positively selected genes. The enrichment of expanded, contracted, and positively selected genes in KEGG pathways was also analyzed using KOBAS.

### 4.5. Identification of Fibrillin Gene Family Members and Phylogenetic Analysis

Fibrillin proteins were identified among the protein sequences of *S. cumingii*’s genome based on a homology search conducted via BLASTP [89] against ten *C. gigas* fibrillin protein sequences obtained from the MolluscDB [90] database. Then, all candidate protein sequences were confirmed on the NCBI Conserved Domain Database (CDD) [91] and Simple Modular Architecture Research Tool (SMART) [92] databases. In addition, the preliminary candidate fibrillin genes were further identified using the hmmsearch tool in HMMER version 3.3.2 [93] with the Hidden Markov Model (HMM) profile of EGF_CA (PF07645). The same workflow was also applied to identify fibrillin protein sequences in *C. gigas*, *C. farreri*, *D. rostriformis*, *O. bimaculoides*, and *P. fucata*. The fibrillin amino acid sequences identified in these four species were aligned using MUSCLE. A phylogenetic tree was constructed using the ML method in IQ-Tree version 1.6.12 [94]. Bootstrapping with 1000 replications was applied to estimate the support rate of branch nodes. The best-fitting model (VT + R7) was selected via ModelFinder in IQ-Tree according to the Bayesian information criterion (BIC). Fibrillins obtained from two species belonging to the phylum Platyhelminthes (*Schistosoma haematobium* and *Schistosoma mansoniwere*) were used as outgroups. The physicochemical parameters, molecular weights (kDa), and isoelectric points (pI) of the fibrillins were calculated in ExPASy version 3.0 [95].

### 4.6. Transcriptome Analysis

Samples of 11 tissues, namely the intestine, foot, hepatopancreas, heart, gonad, adductor, blood, mantle, gill, kidney, and pallial line, were collected from three individuals. Sac tissues were also collected from three individuals at different time points after mantle implantation, i.e., at 3, 6, 12, and 24 h, and at 4, 14, 30, 45, and 60 days. All samples were stored in the RNAstore (TIANGEN, Beijing, China) for transportation. Total mRNA was extracted with TRlzol reagent (Invitrogen, Waltham, MA, USA) according to the manufacturer’s instructions, and three repeat tissues were then mixed for a reverse transcription reaction. Illumina RNA-seq libraries were prepared and sequenced on an Illumina HiSeq2500 platform with a PE150 model. After quality control, high-quality reads were mapped onto the genome of *S. cumingii* using Hisat2 version 2.2.1 [96]. The gene expression level was determined by calculating the transcripts per kilobase of the exon model per million mapped reads (TPM) using Featurecounts (version 2.0.3) [97].

### 4.7. RNA Silencing Assay

Double-stranded RNA (dsRNA) was generated for the RNAi assay via PCR amplification of the *fibrillin* gene under positive selection (Hcu_0920.t1) with a T7 promoter sequence (GGATCCTAATACGACTCACTATAGGG) attached to the primers (forward: GAAAATAATGGTGGATGCG; reverse: CAGAAAAAAGAGCCGATAGT). After PCR amplification, in vitro transcription was conducted using the T7 High Efficiency Transcription Kit (TransGen Biotech, Beijing, China). Phosphate-buffered saline (PBS) and a fragment of the green fluorescence protein (GFP) sequence from the pEGFP-N1 plasmid were used as blank control and negative control dsRNA, respectively. Ten one-year-old mussels from each group were used as the experimental samples. The dsRNA was diluted to 60 μg/100 μL using RNase-free water and was then injected into the adductor. Total RNA from the mantle tissue was extracted 7 days after injection, and qRT-PCR was performed to analyze expression performance. Then, the shell pieces were washed and dried (Figure 4B), and the nacre and prismatic layers of the shell were observed by scanning electron microscopy (SEM).

### 4.8. In Vitro Calcium Carbonate Crystallization Assay

The peptide of the conserved calcium-binding epidermal growth factor domain (EGF-CA) (DIDECAKYASKICQNGKCLNTNPSYTCECYNGYVPDDKNMTCK) was synthesized by Shanghai Qiangyao Biological Co., Ltd. (Shanghai, China) (Appendix A). The peptide was diluted to concentrations of 40 μg/mL and 80 μg/mL before being mixed with a saturated solution of calcium bicarbonate (Ca(HCO_3_)_2_). In the blank control groups, a reaction system consisting of 20 μL Ca(HCO_3_)_2_ solution was used. In the experimental group, the peptides were divided into three concentrations, and 10 μL of different concentrations of peptide was added to 10 μL of Ca(HCO_3_)_2_ solution. The crystallization reaction was carried out on siliconized slides under airtight conditions for 48 h. Crystal morphology was identified by SEM, and crystal types were determined by testing the spectral intensity of crystals using Raman spectroscopy, with Raman shifts ranging from 0 to 4000 cm^−1^.

## 5. Conclusions

In this study, we created a high-quality genome assembly of *S. cumingii* with a size of 2.90 Gb and a contig N50 length of 5.30 Mb. It is the first genome of the Sinohyriopsis genus. A substantial 89.92% of sequences (2.61 Gb) were successfully anchored onto 19 linkage groups. The annotation revealed 37,681 protein-coding genes, and it highlighted the prevalence of repeat elements, explaining why the genome size of *S. cumingii* is so large. A comparative genomics analysis uncovered 752 expanded gene families and 237 genes exhibiting positive selection. Notably, our study provides compelling evidence demonstrating that the remarkable biomineralization ability of this species is driven by the expansion of the *fibrillin* gene. In light of these results, we suggest that freshwater pearl production could be improved using *S. cumingii*. The genome assembly achieved in this study serves as a valuable resource for a wide range of genomic, biological, and ecological investigations into *S. cumingii*. Additionally, it lays the groundwork for practical developments in the freshwater pearl industry, such as molecular breeding and innovations in culture technology. 

## Figures and Tables

**Figure 1 ijms-25-03146-f001:**
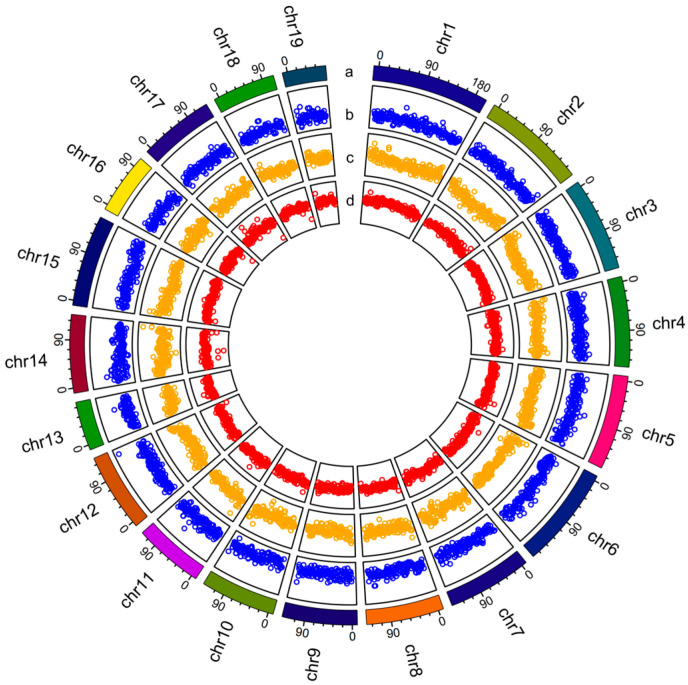
Diagram and genomic landscape of the freshwater pearl mussel *S. cumingii*. From outer to inner circles: a represents the 19 haploid chromosomes at the Mb scale; b represents gene density (blue points) on each chromosome; c represents GC content (orange points) across the genome; and d represents repeat density (red points), drawn in 1 Mb sliding windows.

**Figure 2 ijms-25-03146-f002:**
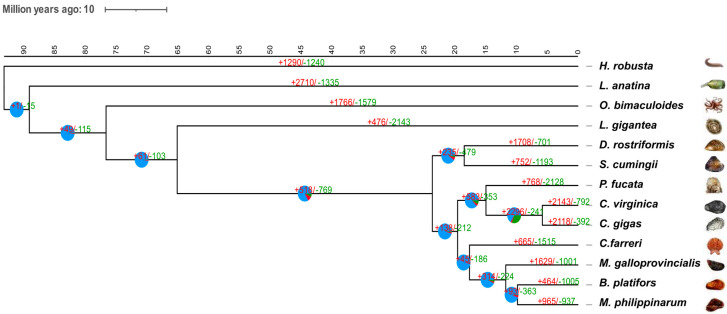
The phylogenetic relationships of *S. cumingii* with other species. The numbers of gene expansion (+) and contraction (−) are shown on the branches. The divergence times are dated and displayed below the phylogenetic tree.

**Figure 3 ijms-25-03146-f003:**
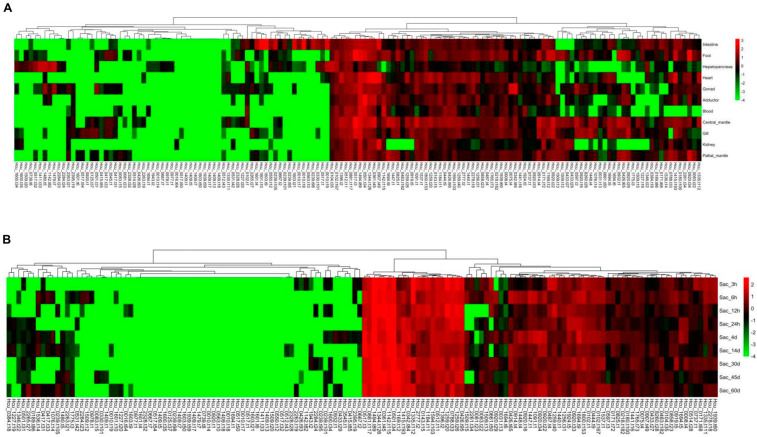
The expression profiles of *fibrillin* genes based on RNA-seq: (**A**) the expression profiles of *fibrillin* genes among eleven different tissues; (**B**) the expression profiles of *fibrillin* genes on sac tissue at different times after mantle implantation.

**Figure 4 ijms-25-03146-f004:**
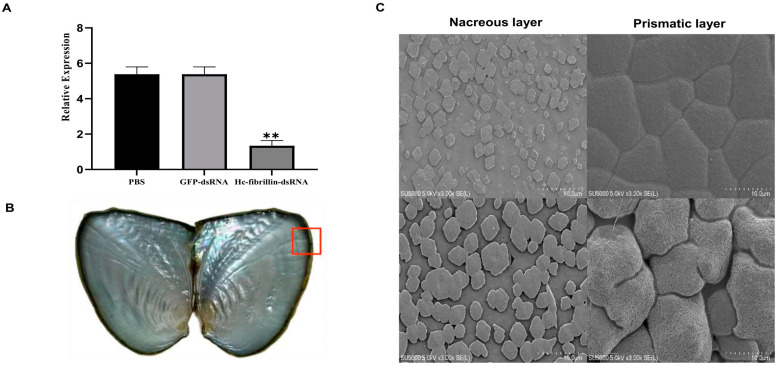
RNA interference analysis of the *fibrillin* gene in the biomineralization of *S. cumingii*: (**A**) Relative expression of the *fibrillin* gene in the mantle after dsRNA injection. Note: “**” indicates a significant difference (*p* < 0.01). (**B**) Inner shell of *S. cumingii* after RNA interference, red box represented location of shell pieces for SEM. (**C**) Microstructure of nacreous and prismatic layers observed after inhibition of the fibrillin gene (bar = 10 μm). Up, PBS and GFP; down, dsRNA-fibrillin.

**Figure 5 ijms-25-03146-f005:**
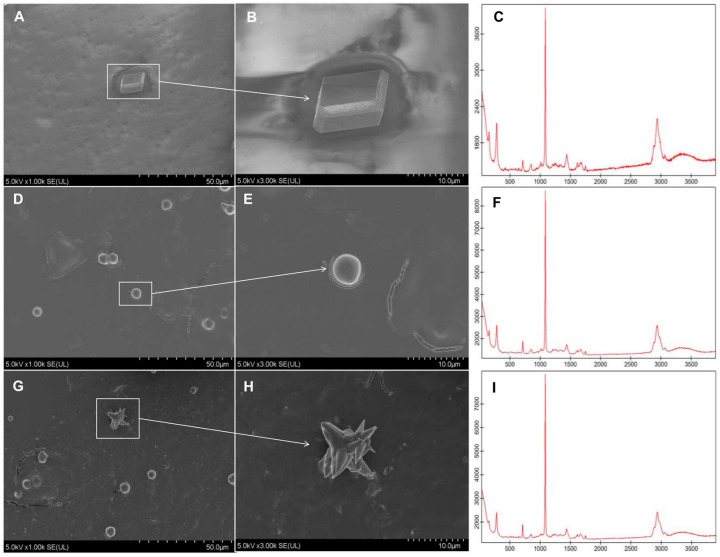
In vitro calcium carbonate crystallization analysis of the EGF-CA peptide: (**A**–**C**) represent SEM images and Raman spectroscopy of saturated Ca(HCO_3_)_2_ solution without peptide, respectively; (**D**–**F**) represent SEM images and Raman spectroscopy of Ca(HCO_3_)_2_ solution with 40 μg/mL EGF-CA peptides, respectively; (**G**–**I**) represent SEM images and Raman spectroscopy of Ca(HCO_3_)_2_ solution with 80 μg/mL EGF-CA peptides, respectively. The *X*-axis of Raman spectroscopy represents the Raman shift, while the *Y*-axis represents the spectral intensity.

**Figure 6 ijms-25-03146-f006:**
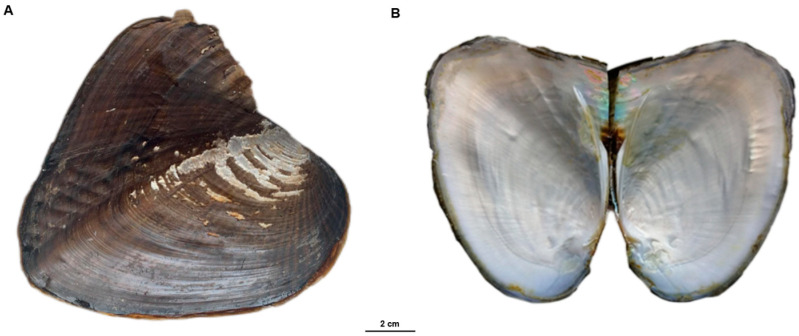
Photograph of a three-year-old triangle sail mussel, *S. cumingii* ((**A**): surface; (**B**): inner shell; bar = 2 cm).

**Table 1 ijms-25-03146-t001:** Statistics of Assembly for *S. cumingii* using Pacbio HIFI data.

Section	Value
Number of contigs	1808
Contig N50 (bp)	5,295,426
Contig N90 (bp)	1,144,919
Total length of contigs (bp)	2,904,942,185

**Table 2 ijms-25-03146-t002:** Statistics of repeat elements for *S. cumingii* assembly.

Element	Number of Elements	Length Occupied (bp)	Percentage
Retroelements	781,650	359,759,912	12.38%
SINEs	0	0	0.00%
Penelope	18,874	7,315,525	0.25%
LINEs	258,394	160,402,819	5.52%
CRE/SLACS	0	0	0.00%
L2/CR1/Rex	83,861	48,684,191	1.68%
R1/LOA/Jockey	38,560	33,760,425	1.16%
R2/R4/NeSL	3247	596,989	0.02%
RTE/Bov-B	69,197	46,843,959	1.61%
L1/CIN4	1401	864,374	0.03%
LTR elements	523,256	199,357,093	6.86%
BEL/Pao	3768	1,264,899	0.04%
Ty1/Copia	22	6808	0.00%
Gypsy/DIRS1	152,335	87,172,551	3.00%
Retroviral	0	0	0.00%
DNA transposons	400,216	383,834,032	13.21%
hobo-Activator	64,261	23,061,802	0.79%
Tc1-IS630-Pogo	34,110	12,365,403	0.43%
En-Spm	0	0	0.00%
MuDR-IS905	0	0	0.00%
PiggyBac	3422	1,122,485	0.04%
Tourist/Harbinger	0	0	0.00%
Other (Mirage, P-element, Transib)	6018	2,447,569	0.08%
Rolling-circles	0	0	0.00%
Unclassified	3,228,829	670,454,699	23.08%
Total interspersed repeats		1,414,048,643	48.68%
Small RNA	0	0	0.00%
Satellites	1	413	0.00%
Simple repeats	922,774	60,734,831	2.09%
Low complexity	54,102	2,669,708	0.09%
Total number of elements	5,387,572	1,477,453,595	100.00%

**Table 3 ijms-25-03146-t003:** Statistics of gene model features for *S. cumingii*.

Section	Results
Genome size (bp)	2,904,942,185
Repeat sequence (bp)	1,477,453,595
Number of genes	37,696
Gene average length (CDS)	1820.50
Gene average length (DNA)	37,988.60
Exon number per gene	6.77
Exon average length (bp)	218.78
Genome GC content (%)	36.07%

## Data Availability

The Whole Genome Shotgun project of *S. cumingii* has been deposited in the NCBI Sequence Read Archive (SRA) database under Bioproject ID: PRJNA909938. The genome assembly data have been deposited at DDBJ/ENA/GenBank under the accession number GCA_028554795.2.

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
