# Peer review of "The First High-Quality Genome Assembly of Freshwater Pearl Mussel *Sinohyriopsis cumingii*: New Insights into Pearl Biomineralization"

_ijms, 2024, doi:10.3390/ijms25063146_

Round 1
Reviewer 1 Report
Comments and Suggestions for Authors
The manuscript reports genome assembly of triangle sail mussel (Hyriopsis cumingii). It needs major revision, including improving of English language by a native speaker or professional editing service. The quality of all figures are poor and needs to be replaced.
Comments on the Quality of English LanguageNeeds revision
Author Response
Dear Reviewers,
Thanks for your letter and for the reviewers’ comments concerning our manuscript entitled “A High-Quality Genome Assembly of Freshwater Pearl Mussel Provides New Insights into the Pearl Biomineralization” (ID: ijms-2836634). The comments are all valuable and greatly assist us in revising and improving our manuscript, as well as providing important guidance for our research. We have carefully considered the comments and made corrections accordingly. We sincerely hope that our revisions are satisfactory and that this latest version of the manuscript will be finally acceptable.
We are most grateful for your advice and look forward to hearing from you at your earliest convenience.
Sincerely yours,
Zhiyi Bai
The main correction in the paper and the responds to the reviewer’s comments are as following:
Response: Thank you for your comments. We have asked professional editors (MDPI English language editing) to improve the English languages of this manuscript. Meanwhile, we changed the figures to make them clearer.

Reviewer 2 Report
Comments and Suggestions for Authors
Dear Author(s),
You can find my comments in the attachment.
Best Regards.

Author Response
Dear Editors and Reviewers,
Thanks for your letter and for the reviewers’ comments concerning our manuscript entitled “A High-Quality Genome Assembly of Freshwater Pearl Mussel Provides New Insights into the Pearl Biomineralization” (ID: ijms-2836634). The comments are all valuable and greatly assist us in revising and improving our manuscript, as well as providing important guidance for our research. We have carefully considered the comments and made corrections accordingly. We sincerely hope that our revisions are satisfactory and that this latest version of the manuscript will be finally acceptable.
We are most grateful for your advice and look forward to hearing from you at your earliest convenience.
Sincerely yours,
Zhiyi Bai
The main correction in the paper and the responds to the reviewer’s comments are as following:
- Comments: Abstract section
- Correcting spacing errors ("high-qualitygenome" to "high-quality genome",
- "bi valves" to "bivalves",
- Improving sentence structure for readability in Abstract section.
- Ensuring subject-verb agreement ("one fibrillin gene play" to "one fibrillin gene plays").
- Removing unnecessary repetition and clarifying the significance of the findings.
Response: Thank you for your comments. We have checked the typos in this section, and revised them.
- Comments: Introduction section
- Provide a citation for the first sentence of the introduction.
- Line 45: What are these challenges ahead for pearl culture technology? Express with literature support.
Responses: Thank you for your comments. The citation for the first sentence has been provided. We also described the challenges ahead for pearl culture technology and expressed with literature support.
- Comments: Results section
- Line 148: Please provide information about the statistical approaches. Which analyses did you perform? In addition, please emphasize the statistical software.
- Please improve the resolution of Figure 3. In addition, please explain Figure 3 in more depth.
- If possible, please provide information on computational time for the statistical analysis.
Response: Thank you for your comments. We have provided information of the statistical approaches with software. Additionally, we have improved the resolution of Figure 3 and explained it in more depth. Unfortunately, we are unable to provide information on computational time for the statistical analysis. Because it has been finished for a long time, resulting that we have forgot how long did us use for computation.
- Comments: Material and Methods section
- Please move this section before the results section.
Response: Thank you for your comments. In our original manuscript submitted to journal, we put this section before the results. But the editors reformatted it and move it after the results and discussion section. We have read the author guidelines, and make sure that the Material and Methods section comes after the results and discussion section.
- Comments: Conclusion section
- Please expand the conclusion section.
Response: Thank you for your comments. We have revised the conclusion section and make them detailed.

Reviewer 3 Report
Comments and Suggestions for Authors
An improved genome assembly of the pearl mussel has been already published. What is new here? The authors completely ignored previous genome assemblies of the pearl mussel. They just looked at themself and did not compare with already publicly available from around the world. See some examples here: https://academic.oup.com/dnaresearch/article/28/2/dsab002/6182681
https://pubmed.ncbi.nlm.nih.gov/37207176/
Figures 1-3: very bad quality of figures are provided.
Figure 4B and 4C: scale bars are clearly visible.
Figure 5: C, F and I are very poor.
Line 290: I do not see a very clear healthy three-year-old triangle sail mussel in Figure 1. It is just merged with genomic landscape. Please provide high quality with scale bar in a separate figure not a merged with genomic landscape.
Line 290, the authors only used one sample to create genome assembly and claimed that a high quality genome. This sample size is very low and not appropriate to provide any solid conclusion of the findings. The authors need to identify what will be the expected effect size between the data supporting or refuting their hypothesis or in other words, the minimum effect size they hope to detect that deviates from the null hypothesis.
I am wondering if the authors have taken ethics permission for performing the experiment.
Author Response
Dear Reviewers,
Thanks for your letter and for the reviewers’ comments concerning our manuscript entitled “A High-Quality Genome Assembly of Freshwater Pearl Mussel Provides New Insights into the Pearl Biomineralization” (ID: ijms-2836634). The comments are all valuable and greatly assist us in revising and improving our manuscript, as well as providing important guidance for our research. We have carefully considered the comments and made corrections accordingly. We sincerely hope that our revisions are satisfactory and that this latest version of the manuscript will be finally acceptable.
We are most grateful for your advice and look forward to hearing from you at your earliest convenience.
Sincerely yours,
Zhiyi Bai
-
- Comments: An improved genome assembly of the pearl mussel has been already published. What is new here? The authors completely ignored previous genome assemblies of the pearl mussel. They just looked at themself and did not compare with already publicly available from around the world.
Response: Thank you for your comments, and sorry for our limited publication review. We have downloaded and read the paper titled “The Crown Pearl: a draft genome assembly of the European freshwater pearl mussel Margaritifera margaritifera (Linnaeus, 1758)”, and the paper was added reference. Since historical times, the inherent human fascination with pearls turned the freshwater pearl mussel M. margaritifera into a highly valuable cultural and economic resource, but pearl harvesting in M. margaritifera is nowadays residual. At present, S. cumingii is the main freshwater pearl mussel in the world. The two species are from different genus. Additionally, we used Pacbio HiFi sequencing to assemble the genome, and finally complete a higher quality genome assembly of S. cumingii which exhibited a greater N50 length than M. margaritifera genome.
- Comments: Figures 1-3: very bad quality of figures are provided. Figure 4B and 4C: scale bars are clearly visible. Figure 5: C, F and I are very poor.
Responses: Thank you for your comments. We have re submitted the Figure 1-3 and Figure 5, and reformatted the Figure 4. They look clearer.
- Comments: Line 290: I do not see a very clear healthy three-year-old triangle sail mussel in Figure 1. It is just merged with genomic landscape. Please provide high quality with scale bar in a separate figure not a merged with genomic landscape.
Response: Thank you for your comments. We have provided a high-quality figure of mussel with scale bar in supplementary Figure.
- Comments: Line 290, the authors only used one sample to create genome assembly and claimed that a high-quality genome. This sample size is very low and not appropriate to provide any solid conclusion of the findings. The authors need to identify what will be the expected effect size between the data supporting or refuting their hypothesis or in other words, the minimum effect size they hope to detect that deviate from the null hypothesis.
Response: Thank you for your comments. Considering the funding of this research, we just use one sample to finish the genome assembly. The BUSCO analysis indicated that the quality of our genome assembly is well. We agreed the importance of the minimum effect size. But if we use multiple samples to finish a better genome assembly, it will cost too much, and we cannot afford it for the moment.
- Comments: I am wondering if the authors have taken ethics permission for performing the experiment.
Response: Generally, mollusks, except Cephalopoda, do not have animal welfare, so we did not show ethical permission for the experiment in the main text.

Reviewer 4 Report
Comments and Suggestions for Authors
Journal: IJMS (ISSN 1422-0067)
Manuscript ID: ijms-2836634
Type: Article
Title: A high quality genome assembly of freshwater pearl mussel provides new insights into the pearl biomineralization
Section: Molecular Genetics and Genomics
In this research, the authors have performed a de novo genome assembly of the freshwater pearl mussel Sinohyriopsis cumingii, a key commercial species for the freshwater pearl culture. Furthermore, they perform comparative genomics, RNA transcriptomics and carbonate crystallization assays, determining the potential role of fibrillin gene in pearl formation in this species. This research is very interesting because this reference genome is a valuable bioinformatic/genomic tool for the scientific community to perform analyses with this species or related ones as congeneric species (~ten species), and the remaining results make this article really comprehensive.
Some information is required for the Material and Methods and Results section, and issues, questions and potential errors along the main text should be solved/clarified. For me, it was impossible to review the Figures due to the low resolution; this must be improved in the next version of the manuscript. Maybe Figures could be uploaded separately in a zipped file in the Submission Menu for authors. I have given a possible solution at the end of this review. I have indicated different suggestions in the main text. The pending issues that need to be addressed before the manuscript can be considered for publication are in the attached PDF in two sections: (1) Content issues and (2) Formatting issues.

Minor editing of English language required.
Author Response
Dear the Reviewer,
Thanks for your letter and for the reviewers’ comments concerning our manuscript entitled “A High-Quality Genome Assembly of Freshwater Pearl Mussel Provides New Insights into the Pearl Biomineralization” (ID: ijms-2836634). The comments are all valuable and greatly assist us in revising and improving our manuscript, as well as providing important guidance for our research. We have carefully considered the comments and made corrections accordingly. We sincerely hope that our revisions are satisfactory and that this latest version of the manuscript will be finally acceptable.
We are most grateful for your advice and look forward to hearing from you at your earliest convenience.
Sincerely yours,
Zhiyi Bai
The main correction in the paper and the responds to the reviewer’s comments are as following:
- Comments: Lines 2-3: I suggest polishing the title, emphasizing that this is the first genome for this species (and for the genus?). Maybe something like this: “The first high-quality genome assembly of freshwater pearl mussel Sinohyriopsis cumingii: New insights into the pearl biomineralization.” I have used Sinohyriopsis cumingii because it is the specific name recognized as valid for the MolluscaBase. Another important reason to me, to change the specific name would be that the name that I am strongly suggesting (Sinohyriopsis cumingii) is the same used for the deposited reference genome in NCBI database. The first time I was looking for this genome, I could not find anything due to use the specific name used in the main text (Hyriopsis cumingii). This problem will be encountered by future readers of this article if the specific name is not changed along the main text. Please change it. This is mandatory for me.
Response: Thank you for your comments. We agreed to emphasize the first genome in this species, and we have revised the title. Additionally, we carefully checked literatures to confirm the scientific names of this species and found many studies use the Sinohyriopsis genus name e. g. (Manuel et al., 2020). Hence, we changed the scientific name in this article so that the readers can find it in the databases quickly, and we will use the novel name in the following studies.
- Comments: Line 36 (after [2]): I recommend adding a sentence to provide more economic context. The value of production in millions of X currency, the number of people working in this sector, or something like that.
Responses: Thank you for your comments. We have added some economic context of pearl production and its economic benefits.
- Comments: Line 38: Why did this species emerge as a favourite species? This should be indicated here.
Response: Thank you for your comments. We have added the indication before this sentence. Because it can produce high quality (big, round and colorful) pearls, this species emerged as a favorite species.
- Comments: Lines 40-43: When discussing quality, what is quality in this context? Maybe the pearl size, shine...? Maybe you could be more concise here. In general, is the quality of freshwater pearls lower than seawater ones? As a curiosity, what is the advantage of freshwater pearls over seawater ones?
Response: Thank you for your comments. The quality of pearls is expressed in size, shape, color and luster et al. We have added this information. Seawater pearls are of better quality than freshwater pearls in some terms, such as luster, while freshwater pearls had thicker nacre. Additionally, freshwater pearls have many colors that seawater ones do not have, for example the dark purple, which is also of high value.
- Comments: Line 45: Additionally, to “genetic breeding” could make sense “to improve aquaculture growing conditions”? If you agree, add it to the main text.
Response: Thank you for your comments. We agreed this point and it has been added to the main text.
- Comments: Line 74: Please change “sequence” to “assembly”.
Response: Thank you for your comments. We have revised it.
- Comments: Line 75: I strongly recommend softening this statement. Please change “presents
evidence” to “showed results” or something similar.
Response: Thank you for your comments. We have revised it.
- Comments: Line 99: What molecular markers? Single Nucleotide Polymorphisms (SNPs)? Please specify in the main text. Are these markers from another research? Please cite it in the main text (Material and Methods section; line 309th).
Response: Thank you for your comments. The molecular markers included simple sequence repeats (SSRs) and single nucleotide polymorphisms (SNPs) from two genetic maps. We have specified in the main text and cited these two researches.
- Comments: Lines 84-91: It could be interesting for the readers to know how much data was removed after the quality filtering steps.
Response: Thank you for your comments. We have added the raw data and clean data of sequences in this section.
- Comments: Line 118 (Table 2): Please add a row at the end of this table with the total values: (i) total number of elements, (ii) total length and (iii) 100%.
Response: Thank you for your comments. We have added this information.
- Comments: Lines 199-209: From my unknown about this species’ aquaculture. Could it make sense to modify/enrich the ion composition of freshwater to change/improve pearl crystallography/quality? In relation to Figure 4C, I have understood that upper Figures (nacreous and prismatic layers) are the best in relation to pearl quality. Did I understand adequately?
Response: In this study, we used the saturated calcium bicarbonate solution for in vivo calcium carbonate crystallization assay to suggest that the balance of peptide concentration is the key to improve the pearl quality. Our previous studies have indicated that the ion composition of freshwater will affect pearl quality. Hence, we just pay attention to the effects of different peptide concentration on crystallography in this study. In the assay of Figure 4, we used RNA interference to verify the biomineralization function of fibrillin gene. The blank control (upper figures) showed the normal structure of shell without any and it might produce high-quality pearls. The RNA silencing group (nether figures) showed that growth of shells was inhibited because of the inhibition of fibrillin gene expression, and it might produce low-quality pearls, even not produce pearls.
- Comments: Line 218: “Female”, please this must be indicated in the Material and Methods section as well.
Response: Thank you for your comments. We have added the indication in Material and Methods section.
- Comments: Line 224: Please include a range between parentheses with the repeat elements proportions in molluscs, for example: “(XX%-ZZ%)”.
Response: Thank you for your comments. We have added the range of repeat elements proportions in molluecs.
- Comments: Line 290: “Healthy” or “phenotypically healthy”? Was it checked the health of the samples in some way (e.g., histologically)? If not, I would recommend using “phenotypically healthy” or something similar.
Response: Thank you for your comments. Strong water spraying is one of the important criteria of health for mussels. We used this method to select the healthy mussel for this study.
- Comments: Line 290: Please change “ cumingii” to “S. cumingii”.
Response: Thank you for your comments. As in our response to Comment 1, we have revised all the scientific names.
- Comments: Line 301: How was the quality of raw data evaluated? Please explain it. How was filtered the raw sequencing data? Please indicate in this paragraph.
Response: Thank you for your comments. We have added the statement of sequences quality control.
- Comments: Line 302: “k-mer”. What software was used for genome-size estimation (e.g., GCE https://github.com/fanagislab/GCE)? Was the genome size calculated as K-mer number/peak depth (Figure S2, Table S2)? Please specify here as well.
Response: Thank you for your comments. We have specified the statement.
- Comments: Line 305: Please specify the type of markers. SNPs? From previous research? If so, then cite it.
Response: Thank you for your comments. We have revised it.
- Comments: Line 322: Please specify the E-value threshold.
Response: Thank you for your comments. We have added the E-value threshold.
- Comments: Lines 325-329: Please change “sets of 10 molluscs and 2 other invertebrate species” to “sets of 10 bivalves and 2 other invertebrate species”. I would put the ten bivalve species first and, in the end, Hellobdella robusta and Octopus bimaculoides. Please change “C. farreri” to “Chlamys farreri”. Please change “C. gigas” to “Crassotrea gigas” and “Crassostrea virginica” to “C. virginica”.
Response: Thank you for your comments. We have revised it. Additionally, considering the abbreviation format of scientific name in author guidelines. Some scientific names are not the first time to use in the main text. Please forgive me for not making changes.
- Comments: Line 346: What method was used? According to Table S12, this would be the Yang-Neilsen method. This would be indicated here.
Response: Thank you for your comments. We have provided additional indications.
- Comments: Lines 399-409: How many samples were used here? I am unsure here.
Response: Each sample had three repeats. But the three repeat tissues were mixed for RNA-seq. We have explained in this section.
- Comments: Line 409: What spectrophotometer model was used?
Response: I am sorry that we did not use spectrophotometer in this study. We just use Raman spectroscopy to identify the crystal types. But we added the test conditions.
- Comments: Lines 411-421: I have missed some examples of direct application: “With these results, freshwater pearl production could be improved using X for Z”. Please add some sentences if you agree. I would also emphasise that it is the first genome of the genus, wouldn't I?
Response: Thank you for your comments. We have revised it in this section.
- Comments: Add the GenBank accession to “GCA_028554795.2”.
Response: Thank you for your comments. We have revised it.
- Comments: Line 17: I strongly recommend changing “dominates” to “leads”. “Dominates” sounds a bit aggressive to me. “Leading” is used in line 35th, for instance.
Response: Thank you for your comments. We agreed to change “dominates” to “leads” and it has been revised.
- Comments: Line 33: I recommend modifying the end of the first sentence. From “lucrative aquaculture industry globally” to “profitable global aquaculture industry”.
Response: Thank you for your comments. We have revised it in this section.
- Comments: Lines 33-34: Remove the first “pearl culture” to “marine and freshwater pearl culture”. Line 72: Please change “imilted” to “limited”. Line 75: Please change “analyssis” to “analysis” or, better “analyses”. Line 104: Please change “The results demonstrated” to “These results showed”.
Response: Thank you for your comments. We have revised them.
- Comments: Line 106 (Figure 1): I strongly recommend splitting this Figure into two. For the first one, please triple its size. There is no page limit in MDPI. In this way, readers can understand your results better.
Response: Thank you for your comments. We have changed the Figure arrangement.
- Comments: Line 125 (Table 3): Please write “cds” and “dna” in capital letters.
Response: Thank you for your comments. We have revised it.
- Comments: Line 128: Please change “found to share” to “being shared”. Change if you agree.
Response: Thank you for your comments. We agreed this change and it has been revised in main text.
- Comments: Line 138: Again, I strongly recommend splitting this Figure into two. For the first one, please quadruple its size. For the second one, please double it. Maybe some figures could be used as supplemental material. Please review it. I am sorry, but I consider that any reader could see Figure 2A adequately, for instance.
Response: Thank you for your comments. We have rearranged the Figure 2.
- Comments: Line 151: This is a small detail. Please be more coherent with the way you write the statistical significance. For instance, in lines 344th and 350th, “P-value” was written. Here, the “p” is in italics, but in line 196th, it is not. All are correct; please select one of them.
Response: Thank you for your comments. They are typos. We have revised them to “p”.
- Comments: Line 169 (Figure 3A): I strongly recommend including this figure as supplemental material with the highest resolution. Figure 3B could be maintained here as Figure 3+X. “X” if you split some previous figures as I recommended above.
Response: Thank you for your comments. We have revised their arrangement.
- Comments: Line 176: Maybe “fibrillin” in italics. Please change it if you agree.
Response: Thank you for your comments. We have revised this typo.
- Comments: Line 337: Perhaps using “replicates” instead of “replications” would be better. Please change it if you agree with it. Line 358: Put “database” before “[89]”.
Response: Thank you for your comments. We agreed and revised it.
- Comments: Lines 445-446: Please put specific names in italics and uniquely put the first letter in capitals →, “Hyriopsis Cumingii” to “Hyriopsis cumingii”. The same for lines 474, 477,483, 491, 494, 497, 506, 509, 514, 519, 522, 528 and so on.
Response: Thank you for your comments. These are typos. We have checked the reference list and revised all of them.
- Comments: Change in the header of all Supplementary Tables “H. cumingii” to “S. cumingii”.
Response: Thank you for your comments. As in our response to Comment 1, we have revised them.
- Comments: If you agree, please “Figure S1” and so on in bold type, the same for Supplementary Tables (“Table S1” in bold type). Tables S6 and S7 were not cited in the main text.
Response: Thank you for your comments. We agreed these changes and they have been revised. However, Table S6 and S7 have been cited.
- Comments: Table S4: What type of marker was used? Please change “marker” to (for instance) “SNPs” or the corresponding marker. Use commas as thousand separators.
Response: Thank you for your comments. We have added the number of SSR and SNP markers which used in this study. The thousand separators have been added in Table S4.
- Comments: Table S8: Please change “L. anatine” to “L. anatina”. Table S12 (header): Please explain the acronym (YN) to Yang-Neilsen method. Change if correct. Table S14 (header): Please explain “pI” in the header.
Response: Thank you for your comments. We have revised these mistake and deficiency in supplementary Table.

Reviewer 5 Report
Comments and Suggestions for Authors
The manuscript ijms-2836634 reports the genomic sequence of Hyriopsis cumingii with particular reference to the genes that contribute to the production of the pearl.
The paper is well outlined and sufficiently clear. There are some typos to correct:
- line 75: correct the word analysis;
- line 77 correct the word family;
- in the titles it is advisable to insert the scientific name in full (chapter 2.1, 2.4, 2.5 and 2.6);
- figure 1 has a low resolution;
- on page 9, on the thirty-sixth line, correct the word fibrillin;
- on page 11, in chapter 4.5 check the third to last line regarding the scientific names;
- in the conclusions, on the fifth line separate the scientific name from the verb.
Apart from these small corrections, the manuscript does not present any particular problems and can be started towards the publication process.
Author Response
Dear Reviewer,
Thanks for your letter and for the reviewers’ comments concerning our manuscript entitled “A High-Quality Genome Assembly of Freshwater Pearl Mussel Provides New Insights into the Pearl Biomineralization” (ID: ijms-2836634). The comments are all valuable and greatly assist us in revising and improving our manuscript, as well as providing important guidance for our research. We have carefully considered the comments and made corrections accordingly. We sincerely hope that our revisions are satisfactory and that this latest version of the manuscript will be finally acceptable.
We are most grateful for your advice and look forward to hearing from you at your earliest convenience.
Sincerely yours,
Zhiyi Bai
The main correction in the paper and the responds to the reviewer’s comments are as following:
- Comments: line 75: correct the word analysis.
Response: Thank you for your comments. We have corrected the typos.
- Comments: line 77 correct the word family
Responses: Thank you for your comments. We have corrected the typos.
- Comments: in the titles it is advisable to insert the scientific name in full (chapter 2.1, 2.4, 2.5 and 2.6);
Response: Thank you for your comments. We have checked the scientific name. Because they are not the first time for use, the genus was abbreviated to the first letter.
- Comments: figure 1 has a low resolution
Response: Thank you for your comments. We have re submitted all the and make them clearer.
- Comments: on page 9, on the thirty-sixth line, correct the word fibrillin.
Response: Thank you for your comments. We have corrected the word.
- Comments: on page 11, in chapter 4.5 check the third to last line regarding the scientific names.
Response: Thank you for your comments. We have checked the scientific names. As mentioned above, they were not the first time for use in manuscript. Hence, we used the abbreviated form.
- Comments: in the conclusions, on the fifth line separate the scientific name from the verb.
Response: Thank you for your comments. We have revised this typo.

Round 2
Reviewer 1 Report
Comments and Suggestions for Authors
The requested changes has been performed. The quality of English language and figures resolution has been improved
Reviewer 2 Report
Comments and Suggestions for Authors
The authors have made the corrections indicated. The article can be published in its final version.
Reviewer 3 Report
Comments and Suggestions for Authors
The authors have made all the necessary edits. The manuscript now appears much better readable.